# The Impact of Medical Nutrition Intervention on the Management of Hyperphosphatemia in Hemodialysis Patients with Stage 5 Chronic Kidney Disease: A Case Series

**DOI:** 10.3390/ijerph20065049

**Published:** 2023-03-13

**Authors:** Elena Moroșan, Violeta Popovici, Viviana Elian, Adriana Maria Dărăban, Andreea Ioana Rusu, Monica Licu, Magdalena Mititelu, Oana Karampelas

**Affiliations:** 1Department of Clinical Laboratory and Food Safety, Faculty of Pharmacy, “Carol Davila” University of Medicine and Pharmacy, 6 Traian Vuia Street, 020945 Bucharest, Romania; 2Department of Microbiology and Immunology, Faculty of Dental Medicine, Ovidius University of Constanta, 7 Ilarie Voronca Street, 900684 Constanta, Romania; 3Department of Diabetes, Nutrition and Metabolic Diseases, “Carol Davila” University of Medicine and Pharmacy, 8 Eroii Sanitari Blvd, 050471 Bucharest, Romania; 4Department of Diabetes, Nutrition and Metabolic Diseases, “Prof. Dr. N. C. Paulescu” National Institute of Diabetes, Nutrition and Metabolic Diseases, 030167 Bucharest, Romania; 5Faculty of Pharmacy, “Vasile Goldiș” Western University of Arad, 86 Liviu Rebreanu Street, 310045 Arad, Romania,; 6Department of Medical Psychology, Faculty of Medicine, “Carol Davila” University of Medicine and Pharmacy, 8 Eroii Sanitari Blvd, 050474 Bucharest, Romania; 7Department of Pharmaceutical Technology and Biopharmacy, Faculty of Pharmacy, “Carol Davila” University of Medicine and Pharmacy, 6 Traian Vuia Street, 020945 Bucharest, Romania

**Keywords:** chronic kidney disease, hemodialysis, phosphatemia, medical nutrition therapy, phosphate binder drugs, compliance

## Abstract

The treatment and interdisciplinary management of patients with chronic kidney disease (CKD) continue to improve long-term outcomes. The medical nutrition intervention’s role is to establish a healthy diet plan for kidney protection, reach blood pressure and blood glucose goals, and prevent or delay health problems caused by kidney disease. Our study aims to report the effects of medical nutrition therapy—substituting foods rich in phosphorus-containing additives with ones low in phosphates content on phosphatemia and phosphate binders drug prescription in stage 5 CKD patients with hemodialysis. Thus, 18 adults with high phosphatemia levels (over 5.5 mg/dL) were monitored at a single center. Everyone received standard personalized diets to replace processed foods with phosphorus additives according to their comorbidities and treatment with prosphate binder drugs. Clinical laboratory data, including dialysis protocol, calcemia, and phosphatemia, were evaluated at the beginning of the study, after 30 and 60 days. A food survey was assessed at baseline and after 60 days. The results did not show significant differences between serum phosphate levels between the first and second measurements; thus, the phosphate binders’ initial doses did not change. After 2 months, phosphate levels decreased considerably (from 7.322 mg/dL to 5.368 mg/dL); therefore, phosphate binder doses were diminished. In conclusion, medical nutrition intervention in patients with hemodialysis significantly reduced serum phosphate concentrations after 60 days. Restricting the intake of processed foods containing phosphorus additives—in particularized diets adapted to each patient’s comorbidities—and receiving phosphate binders represented substantial steps to decrease phosphatemia levels. The best results were significantly associated with life expectancy; at the same time, they showed a negative correlation with the dialysis period and participants’ age.

## 1. Introduction

According to the World Health Organization, chronic kidney disease (CKD) is a progressive loss of kidney function [1]. The International Organization Kidney Disease: Improving Global Outcomes (KDIGO) established the criteria for CKD as the glomerular filtration rate (GFR) < 60 mL/min per 1.73 m^2^ for >3 months [2]. 

The more recent data from the National Institute of Diabetes and Digestive and Kidney Diseases (NIDDK) [3] show that diabetes and high blood pressure are the two most common causes of CKD. The CKD incidence is slightly more common in women (14%) than men (12%). People aged 65 or older are most affected (38%); the ones in the age range of 45 to 64 are the second regarding CKD relevance (12%), followed by those ages between 18 and 44 years (6%) [4]. It has a progressive evolution into kidney failure—GFR < 15 mL/min per 1.73 m^2^ or treatment by dialysis—affecting more than 10% of people worldwide [5]. Kidney failure—known as the end stage of kidney disease (ESKD), diminishes the quality of life and causes premature death, meaningfully associated with cardiovascular diseases [6]. The treatment consists of kidney replacement therapy (KRT): dialysis (hemodialysis, peritoneal dialysis, hemofiltration, hemodiafiltration), and kidney transplant [7]. 

On the other hand, CKD treatment and interdisciplinary approaches continue improving long-term patient outcomes [8]. Thus, a CKD patient’s healthcare team must include a registered dietician as an essential medical professional. His or her role is to establish a healthy diet plan for kidney protection, reach blood pressure and blood glucose goals [8], and prevent or delay health problems caused by kidney disease [9]. This food and nutrition specialist performs a medical nutrition therapy (MNT) [10], which can help slow CKD progression, prevent or treat complications and improve quality of life [11]. Therefore, CKD patients can protect their bones and blood vessels by limiting fluids [12], eating a low-protein diet [13,14,15], and diminishing the intake of salt [16], potassium [17], phosphorus [18], and other electrolytes. 

When the blood phosphate level is high [19], CKD affects calcium-phosphate metabolism and bone homeostasis, inducing osteomalacia, renal osteodystrophy, and vascular and ectopic calcification, resulting in chronic kidney disease mineral bone disorder (CKD MBD) [18,20]. Moreover, various experimental studies highlighted the mechanisms by which phosphate in excess may adversely affect the cardiovascular system [21,22]. Thus, it may directly contribute to vascular damage by pro-inflammatory actions on the vascular smooth muscle cell leading to endothelial dysfunction and promoting vascular calcification [23,24]. Furthermore, a high dietary phosphate content may contribute to atherogenesis [25] and has also been linked to a more rapid progression of CKD to ESKD [19,26,27]. Using data from the US Renal Data System, Block et al. [4] found an increased risk of death (relative risk, 1.27) associated with serum phosphate levels > 6.5 mg/dL. In addition, excess phosphates could inhibit the renal transformation of 25(OH) vitamin D to 1.25(OH)_2_ vitamin D, leading to fibroblast growth factor 23 (FGF23) [28] and parathyroid hormone (PTH) [29] hypersecretion.

In current clinical practice, the management of hyperphosphatemia is based on four main strategies: (i) restriction of dietary phosphate intake [30,31]; (ii) reduction of its intestinal absorption; (iii) phosphate removal by dialysis; and (iv) treatment and prevention of renal osteodystrophy. Previous studies have proven that managing eating behavior through MNT determines a favorable evolution of hyperphosphatemia in dialysis patients and a decrease in the treatment dose with phosphate binders. 

Our study aims to demonstrate the impact of a diet low in processed foods [32] with a low content of inorganic phosphates [33] and phosphate binder drug administration on phosphatemia and implicitly on the prognosis of stage 5 CKD patients with hemodialysis. To achieve this, we used qualitative methods to explore the CKD patients’ adherence to MNT and quantitative ones—corresponding laboratory parameter measurements and analyzing their evolution.

## 2. Methods

### 2.1. Study Design

Our clinical research is conducted as a case series [34,35], a descriptive study of stage 5 CKD patients on hemodialysis [36], included in an MNT program with low inorganic phosphate content for 2 months. The aim is to observe the impact of a low-phosphorus diet associated with phosphate binders’ agents on phosphatemia. The Romanian Agency of Medicines and Medical Devices approved our study by Authorization for Conducting Clinical Studies with Therapeutic Benefit 171/03.11.2019.

### 2.2. Descriptive Analysis of the Patients’ Series

Patients from a tertiary center in Bucharest with stage 5 CKD treated by hemodialysis and having high serum phosphates levels (>5.5 mg/dL) were included in this MNT program. 

#### 2.2.1. Inclusion criteria

The selected series contains patients of both sexes, aged over >18 years, with stage 5 CKD on hemodialysis treatment for at least 12 months and constant serum phosphorus levels >5.5 mg/dL during the previous 3 months. In addition, they should have good cognitive function, and the ability to read and write is required.

#### 2.2.2. Exclusion criteria

The following patients have been excluded: those with early CKD and PTH > 1500 pg/mL, with infectious diseases, enteral or parenteral therapy, malabsorption, and cognitive or physical limitations.

#### 2.2.3. Details

Forty individuals were initially planned, but only eighteen (45%) accepted the protocol and consented to rigorously respect all the study phases.

Each participant had three study visits. 

The intact PTH levels were measured for the first time to assess the inclusion criteria. The monitored parameters before the MNT initiation were established during the first one: Kt/V, calcemia, and phosphatemia. According to their values, the treatment with phosphate binders was prescribed upon guideline recommendations [37]. 

Pending MNT, each participant completed a survey regarding food preferences and eating behavior.

After 30 days, the second evaluation consisted of laboratory findings, measuring the levels of previously mentioned parameters. According to their values, the medication should be adjusted. 

After 60 days from the beginning, the third evaluation qualitatively investigated the diet and compliance with the nutrition changes through a survey. The levels of all parameters were also measured, and the drug prescription was modified.

### 2.3. Medical Nutrition Intervention

All 18 patients included in this study performed the following procedures:Completing the questionnaire regarding food preferences and their consumption frequency at each of the three study visits.They received an instruction to edit a food diary structured for seven days regarding the type of food and liquids ingested, the frequency of consumption, and details about gastroenterology where applicable. The patient had to write down the meals immediately after consumption to reduce the risk of omissions.The food diary was analyzed for each food category, thus obtaining a profile of the patient’s eating habits before the start of the nutrition intervention. From this analysis, the parameters of the frequency of consumption of specific food categories resulted.

After analyzing the data obtained in the previously mentioned stages, the nutrition intervention plan was presented to the patients. They received written instructions regarding the allowed and recommended foods and those that should be avoided daily. They were also informed about gastro-technics adapted to the food condition. These aspects were adapted monthly depending on the measurements of specific biochemical parameters.

The medical nutrition intervention consisted of the following:The nutritionist recommended three meals daily and two snacks in 2 h, customized according to patients’ schedules and comorbidities.All patients were advised not to consume foods with a high phosphorus, potassium, or sodium content for 4 h before collecting samples for specific analyses.Foods containing additives and mainly inorganic phosphates and which are generally processed: processed and matured dairy products, acidic carbonated drinks, frozen bakery and pastry products, processed meat and sausages, canned fish or meat, frozen doughs, processed sweets, chocolate and products with a high cocoa content, oilseeds, and fruits, dried legumes, were eliminated from the diet.The instructions related to gastro-technics were the following: preparing food mainly by boiling, increasing the contact surface/permeability of the food in the immersion liquid, repeating the boiling process depending on the category of processed food, and avoiding extremely high temperatures in the cooking process.Recommendations consisted of increasing the proportion of foods of natural origin, unprocessed, containing phosphorus, potassium, and organic calcium: fresh meat prepared at home, freshwater fish, fresh products with low-fat content, eggwhite, sweets prepared at home, bread made from white flour prepared at home or with a reduced range of additives, cooked vegetables, and roots, olive oil, and vegetable fats, fruits with a low potassium content.Recommendations were made for consulting food products’ labels and avoiding consuming those with additives and/or preservatives.Personalized advice was focused on water intake according to BCM measurements to avoid hyperhydration.Personalized treatment with phosphate binder drugs was prescribed.

The patients received a food diary in which the food consumed during a week was registered and presented at the next visit. To adjust further recommendations, the diet composition was analyzed and corroborated with the measured parameters (Kt/V, phosphatemia, and calcemia).

### 2.4. Clinical Laboratory Analyses

Parathyroid hormone—commonly measured once every 3 or 6 months—was evaluated in this first step, aiming to prescribe suitable phosphate binders because it ensures the regulation of the calcium and phosphorus distribution in the body. Intact PTH level was measured using an immunoradiometric assay [38,39,40].

A colorimetric method was performed to assess serum calcium levels using 1,8-Dihydroxy-3,6-disulpho-2,7-naphthalene-bis (azo)-dibenzenearsonic acid (Arsenazo III) at neutral pH, as previously described [41]. 

Serum phosphorus levels were quantified through spectrophotometry using ammonium molybdate in an acid medium [42].

The Kt/V indicator is investigated monthly in the case of hemodialysis patients. Its variation from one evaluation to another represents the value of the urea clearance (Kt) normalized for the urea distribution volume (V) [43]; more precisely, it is a measure of what dose of dialysis is administered [44], measured through a standard method, previously displayed [45].

### 2.5. Data Analysis

Statistical Package for the Social Sciences (IBM SPSS 20) [46] was used for statistical analysis and graphical representations.

Data analysis was carried out, considering all three measurements. Data was provided by the clinic respecting the terms of confidentiality: age, sex, height, weight, dialysis period, vascular access route, PTH, Kt/V level, phosphatemia, calcemia, phosphate binder treatment, and food preferences.

Friedman and Wilcoxon tests and the descriptive analysis of the variables (mean, standard deviation, minimum and maximum) were used.

We applied the Friedman test because the samples (or variables) are pairs, each subject being tested at least three times (three measurements for Kt/V level, serum phosphate, serum calcium, phosphate binder administration); the dependent variable (DV) is quantitative or ordinal. The Wilcoxon test first calculates the difference between the scores of the two variables for each subject (after, before), establishes the sign for each difference, and finally, ranks the differences in absolute value. The values around zero will be ignored because they do not provide information.

Effect size (*r*) of the Wilcoxon test.

The effect size was calculated using Equation (1):(1)$$r=\sqrt{\frac{z^{2}}{n}}$$
where *z* is the Wilcoxon test result, and *n* is the number of individuals.

The analysis of variables, respectively, the correlation between phosphatemia and all other ones, was performed through principal component analysis, using XLSTAT 2022.2.1. by Addinsoft (New York, NY, USA).

## 3. Results and Discussion

Eighteen stage 5 CKD patients on hemodialysis (50% women and 50% men) were enrolled in the study with an age range of 35–73 years and a mean age of 58.22 ± 10.719 years. Their median height is 170.94 ± 6.584 cm, and their mean dry weight is 75.25 ± 22.446 kg. Their dialysis period belongs to a large domain of 2–15 years, with a mean of 6.33 ± 4159 years (Table 1). 

The hemodialysis period is essential in our descriptive analysis, influencing patient compliance. Therefore, more receptivity and ability to understand the recommendations were observed in patients with shorter dialysis periods than those with more than 5 years—the resistance to guidance increased due to unwanted trial experiences and low results obtained. A recently published crossectional study [47] reports that of 800 eligible patients, only 497 consented and completed all assessments. The remaining 303 eligible patients refused the agreement and/or were reluctant to research procedures because they participated in many research studies [47] without substantial results. In another clinical trial, of 56 estimated participants, only 32 agreed to participate and completed their interview (57%) [48].

Parathyroid hormone (PTH)—commonly measured once every 3 or 6 months—was evaluated in this first step to suitably prescribe the phosphate binders because it ensures the regulation of the calcium and phosphorus distribution in the body. The PTH level range was 30–1393 pg/mL, with a mean of 371.89 pg/mL and an SD of 366.45. Serum-intact PTH values between 100 and 300 pg/mL do not predict the degree of bone turnover in dialysis patients (8 of the 18 patients have PTH values > 300 pg/mL). 

The patients enrolled in the study also present the following comorbidities: diabetes type I (3), diabetes type II (2), arterial hypertension stage II (6), chronic HCV hepatitis with cirrhosis (2), heart failure (3), and neoplasms (2). The dietary recommendations considered all these associated conditions.

The access path for dialysis is an arteriovenous fistula (AFV, 67% of patients) and a central venous catheter (CVC) for the remaining 33%. Most patients have AVF accesses over 12 weeks old with few reparative interventions, making them superior to CVC vascular prostheses.

The initial food survey displays the patients’ food preferences, marked in the increasing order of consumption frequency. Thus, the following notations are available: 1 = very rarely—once per month, 2 = occasionally—once per week, 3 = rarely—once to every few days, 4 = often—once per day, and 5 = very often—several times per day (Table 2).

The parameters with three measurements are displayed in Table 3.

The Kt/V values do not report a significant variation from month to month, which shows us, depending on the mean value, that the dialysis dose had a slight decrease.

The evolution of the calcemia registers an increase in the mean value in the second evaluation, reaching the value of 9.23 mg/dL. The third evaluation reaches approximately the mean value of the first one. The mean values recorded in all measurements fall within the recommended reference range of 8.5 mg/dL–10.2 mg/dL. 

The results were statistically analyzed using Friedman and Wilcoxon tests and presented in Appendix A.

After 2 months, the ordered foods ingested by patients with increased frequency changed compared to the initial survey. Foods with significant phosphorus intake rank lower, indicating that the nutrition recommendations have been followed (Table 2). Moreover, some foods with high phosphate content (processed dried vegetables and spice blends, canned meat, pasta, oil seeds (nuts, almonds, peanuts), canned fish, and whole grains) were substituted by other ones more suitable according to MNT (white and red meat boiled, potatoes, eggwhite, rice, cream cheese).

The phosphatemia did not have significantly different values in the first month of MNT. The favorable evolution of the serum phosphate level following the medical nutrition intervention was obtained after 2 months for 16 of the 18 patients. The nonparametric tests Friedman and Wilcoxon (Appendix A) reported significant differences between all three measurements (Friedman test results show χ^2^ = 16.778, *p* < 0.001 for 18 patients). Negative ranks are found for 16 of the 18 subjects; this means a decrease in the level of phosphate measured in the third month compared to the measurement in the first month on 16 patients. The two positive ranks show an increase in the serum phosphate level compared to the first month in two patients (Appendix A).

The positive correlation between the phosphatemia level and restricted foods’ frequency of consumption, dialysis period, age, calcemia, and Kt/V is displayed in Figure 1. 

Thus, before MNT, at the first evaluation (Figure 1A), phosphatemia is substantially correlated with the dialysis period (*r* = 0.999, *p* < 0.05) and PTH value (*r* = 0.998, *p* < 0.05). It is strongly positively correlated with the age, Kt/V, and restricted foods frequency of consumption *r* = [0.878–0.994], *p* > 0.05 (Figure 1A and Appendix A). 

On the final evaluation, after 2 months of MNT, the high positive correlation between phosphatemia and the other parameters (dialysis period, consumption frequency of cheese/yogurt, snacks, fast food, chocolate, pastry, processed dried legumes) is more statistically significant: *r* = 0.999; *p* < 0.05 (Figure 1B).

Moreover, all previous observations were supported by the places of different phosphate levels in the correlations biplots from Figure 1 (A and B).

A general PCA correlation biplot between variable parameters in both phases is displayed in Figure 2. The two principal components explain 96.92% of total data variance, with 88.21% attributed to the first (PC1) and 8.71% to the second (PC2). PC1 is associated with all variables.

Phosphates level also reports a good correlation with the frequency of consumption of processed cheeses and snacks (*r* = 0.799–0.805, *p* > 0.05). 

Available data of the entire study show a statistically significant high positive correlation between phosphatemia and the greatest part of the parameter evaluated (intake frequency of various restricted foods, age, and dialysis period): *r* = [0.883–0.967], *p* < 0.05 (PCA Correlation Matrix from the Appendix A). 

De Fornasari et al. obtained similar results in 67 participants monitored for 90 days [49]. The phosphatemia level significantly decreased (from 7.2 to 5.0 mg/dL); the reported values are similar to those in Table 3. Calcemia and Kt/V values reported insignificant changes in the final evaluation.

Two phosphate binder drugs and one food supplement were prescribed to decrease serum phosphate levels (Table 4). 

Renagel 800 mg (Sanofi, Berkshire, UK) contains Sevelamer hydrochloride [50]. OsvaRen 435/235mg (Vifor Fresenius Medical Care Renal Pharma UK Ltd.) has calcium acetate 435 mg, equivalent to 110 mg of calcium, and magnesium carbonate, heavy 235 mg, equivalent to 60 mg of magnesium [51,52]. Prodial (Bioeel^®^, Targu Mures, Romania) is a food supplement with a calcium/magnesium composition similar to OsvaRen [53]. 

For 60 days, the phosphate binder treatment did not change. Thus, for two months, three patients followed treatment with OsvaRen, six tablets daily, eleven patients with Prodial, also six tablets daily, and four received Renagel, six tablets daily (Table 4).

In the final evaluation, the MNT intervention showed its efficacy by adjusting the phosphate-chelator treatment with lower doses (from six tablets per day to five, four, and three tablets per day). There were significant differences between the recommended phosphate binder doses in the three months [χ^2^ = 26; *p* < 0.001], according to Friedman and Wilcoxon tests (Appendix A). For 13 patients, the phosphate-chelator dose was decreased in the third month (13 negative ranks). For five, the doses remained the same, so the result of the Wilcoxon test (Appendix A) shows a significant difference between both recommended doses (the third evaluation versus the first one). 

One-third of monitored patients (33%) had their daily dose decreased from six tablets of Prodial per day to five tablets per day. The most significant dose diminution was recorded in 11% of patients, from six Prodial tablets per day to three tablets (Table 4).

Data analysis allows for a global image regarding compliance with the MNT of participants. This aspect varies inversely proportionally to the age of individuals and the dialysis period; a reason supporting this observation could be a loss of health-being hope due to a long period of failed trials and associated comorbidities.

Numerous clinical studies reported a low adherence of participants to phosphate binders treatment [54,55,56]. Khor et al. showed that phosphate binders—mainly calcium-based (92.9%)–were prescribed to 98.2% of patients, but only 59.5% adhered to the phosphate binder drug prescription [47]. Another research team highlighted the effect of individual health education—conducted by a physician with short-term nutrition training—on hyperphosphatemia and adherence to MNT [57]. Moreover, a recent clinical trial reported that management of hyperphosphatemia can be achieved through a mobile application (similar to dietitian management), with the additional benefit of titrating phosphate binder doses according to individual meal choice [58].

Performing a multicenter cross-sectional study, Kurita et al. [28] highlighted the positive relationship of the CKD stage with health expectation, evidencing hope’s essential role in the psychological and physiological manifestations of adherence to the treatment.

On the other hand, a recent qualitative study including patients with CKD in stages 2–4, conducted by Rivera et al. [48], reported multiple factors correlated with treatment adherence. First, patient factors, which include numerous comorbidities, chronic diseases, motivation, and outlook. Second, healthcare team factors: attentiveness, care, availability/accessibility, empathy, and communication style. Third, treatment planning factors: lack of a clear plan, proactive and dynamic research, provider-focused treatment objectives, and shared decision-making. As a result, the response of a CKD patient could be very different: positive feedback, lack of information, perceived capability deficit, and disagreement with treatment [48]. 

The present study was conceived as a case series (also known as clinical series) [35], one of the most common types of medical study designs that describe the experience of a small group of people [59]. Because it did not have a control group or any form of randomization employed, our clinical study could be considered relatively efficient and cost-saving [60]. This study’s results are closer to those obtained in routine clinical practice and may be more relevant than a randomized trial [61]. The external validity [36] is also appreciable in our case series because we included a diverse patient range (with different ages, comorbidities, and other particularities). Thus, our results could be applied to clinical practice in other centers.

However, the lack of a control group is the main limitation of our study; thus, we cannot establish whether the outcomes are exclusively the effect of MNT or whether other variables cause them. Moreover, our clinical series was not a randomized or double-blind study; therefore, a rigorously established protocol is missing, and data collection is incomplete, leading to bias concerns. For example, the amount of phosphate intake could not be quantified because the patients prepared their meals at home, and the foods had different provenance; hence, it was impossible to do their biochemical analysis. Finally, this short-term clinical study lasted only 2 months. All these previously mentioned aspects limit our study’s generalizability. Further research will confirm these results with extensive groups of patients in more extended periods, following suitable protocols.

## 4. Conclusions

Following a low phosphorus diet during medical nutrition intervention, our stage 5 CKD patients on hemodialysis revealed good outcomes: their serum phosphorus levels decreased, thus diminishing the received phosphate binders’ doses after 60 days. 

Our study shows that restricting the intake of processed foods containing phosphate additives—by individualized diets according to each patient’s comorbidities—associated with phosphate binder drug administration is essential to decrease phosphatemia levels. 

The best results were positively correlated with health-related expectancy and negatively correlated with the dialysis period and the age of participants.

## Figures and Tables

**Figure 1 ijerph-20-05049-f001:**
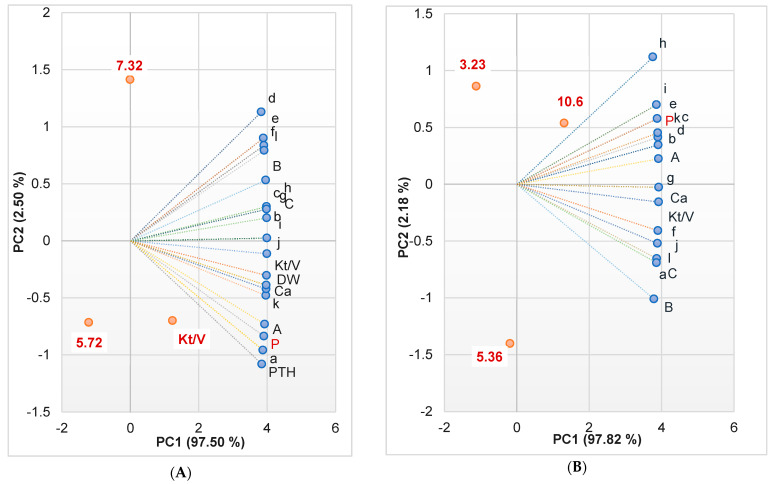
PCA Correlation biplot between phosphate levels (P) and other variable parameters: frequency of consumption of food rich in phosphates in the initial phase (**A**) and final phase (**B**) a—sausages, b—processed cheeses, c—snacks, d—fast food, e—chocolate, f—cola, beer, g—pate, almond butter, h—milk, yogurt, i—pastry, j—oil seeds, k—dried legumes, l—bread, and patients’ data (A—dialysis period, B—age, C—high, Ca—calcemia, Kt/V—dialysis dose).

**Figure 2 ijerph-20-05049-f002:**
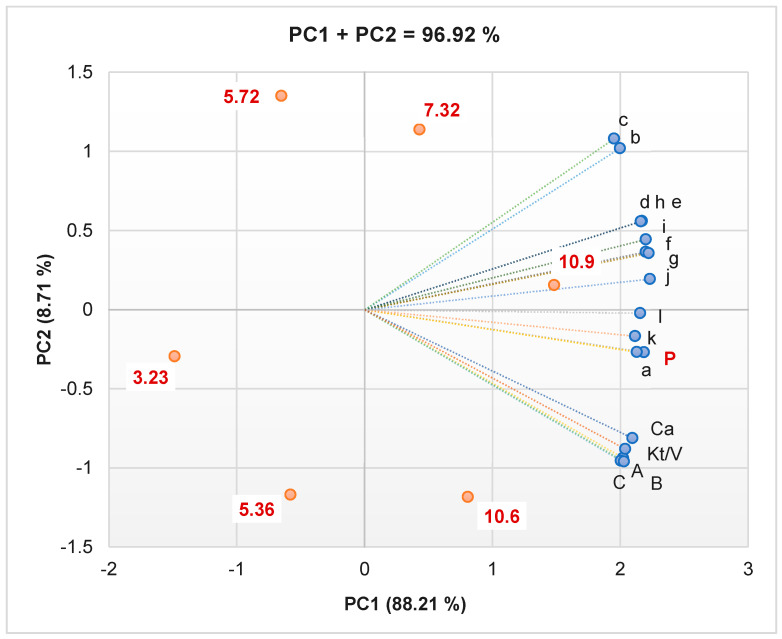
PCA correlation biplot between phosphate levels (P) and other variable parameters: frequency of consumption of food rich in phosphates during the study: a—sausages, b—processed cheeses, c—snacks, d—fast food, e—chocolate, f—cola, beer, g—pate, almond butter, h—milk, yogurt, i—pastry, j—oil seeds, k—dried legumes, l—bread, and patient data: A—dialysis period, B—age, C—high, Ca—calcemia, Kt/V—dialysis dose.

**Table 1 ijerph-20-05049-t001:** Descriptive analysis of participants.

Patients	N	Minim	Maxim	Mean	SD
Age (years)	18	35	73	58.22	10.719
Height (cm)		158	182	170.94	6.584
Dry weight (kg)		40	126	75.25	22.446
Dialysis period (years)		2	15	6.33	4.159
PTH value (pg/mL)		30	1393	371.89	366.446
*Other particularities*					
*Sex*		*N*		*%*	
F		9		50	
M		9		50	
*Comorbidities*					
Diabetes I		3		16.66	
Diabetes II		2		11.11	
Arterial hypertension stage II		6		33.33	
Chronic HCV hepatitis + cirrhosis		2		11.11	
Heart failure		3		16.66	
Neoplasms		2		11.11	
*Access way of dialysis*					
AVF		12		67%	
CVC		6		33%	

N—number of individuals, %—percent, the range of numerical values: (min–max); SD—standard deviation. AFV—arteriovenous fistula; CVC—central venous catheter.

**Table 2 ijerph-20-05049-t002:** Food preferences and frequency of consumption before and after medical nutrition intervention.

Foods		Frequency of Consumption							
		Min	Max	Mean	SD	Min	Max	Mean	SD
		Before MNT				After 2 Months of a Renal Diet			
*Same foods*									
Bread		4	5	4.67	0.485	3	5	4.06	0.639
Sausages/processed meat		4	5	4.28	0.461	1	3	1.61	0.698
Processed cheese.		3	5	4.00	0.594	1	2	1.11	0.323
Salty snacks/chips		3	4	3.56	0.511	1	3	1.56	0.705
Fast-food		2	4	3.50	0.618	1	3	1.61	0.608
Chocolate		2	4	3.39	0.698	1	3	1.5	0.618
Cola, beer		2	4	3.33	0.686	1	3	2.0	0.594
Pâté/Peanut butter		2	4	3.11	0.583	1	3	1.78	0.732
Milk/yogurt		2	4	3.11	0.676	1	3	1.22	0.428
Pastry		2	4	3.00	0.686	1	3	1.44	0.616
Oilseeds		2	4	2.94	0.639	1	3	2.00	0.594
Dried legumes		1	3	1.78	0.548	1	2	1.28	0.461
Before MNT						After 2 months of a renal diet			
Min	Max	Mean		SD		Min	Max	Mean	SD
*Different foods*									
Processed legume and spice blends						White meat			
2	3	2.78		0.428		3	4	3.56	0.511
Canned meat						Boiled red meat			
1	3	2.33		0.840		2	3	2.44	0.511
Pasta						Potatoes			
1	3	2.06		0.416		2	4	3.44	0.705
Nuts, almonds, peanuts						Eggwhite			
1	3	1.78		0.548		2	3	2.78	0.428
Canned fish						Fresh fish			
1	3	1.72		0.669		1	3	2.22	0.808
Whole grains						Rice			
1	2	1.22		0.428		2	3	2.44	0.511
Corn						Diary snacks			
1	2	1.17		0.383		2	4	3	0.594

Frequency of consumption: 1 = very rarely—once per month, 2 = occasionally—once per week, 3 = rarely—once to every few days, 4 = often—once per day, and 5 = very often—several times per day; the range of frequency values: (min–max); SD—standard deviation; N—patients’ number; MNT—medical nutrition therapy.

**Table 3 ijerph-20-05049-t003:** Kt/V, calcemia, and phosphatemia values.

Measurement	Min	Max	Mean	SD
*Kt/V*				
1	1.20	1.76	1.445	0.135
2	1.23	1.70	1.445	0.104
3	1.25	1.65	1.441	0.087
*Calcemia (mg/dL)*				
1	8.24	9.93	8.921	0.482
2	8.15	10.02	9.230	0.516
3	8.03	9.94	8.834	0.631
*Phosphatemia (mg/dL)*				
1	5.72	10.90	7.322	1.453
2	5.41	10.80	6.877	1.412
3	3.23	10.60	5.368	1.797

Measurements: 1—initial evaluation before MNT; 2—after 30 days; 3—after 60 days in the final evaluation. The range of frequency values: (min–max); SD—standard deviation; N—patients’ number; MNT—medical nutrition therapy; Kt/V—dialysis dose.

**Table 4 ijerph-20-05049-t004:** Phosphate binder agents prescribed after the first and second evaluations (A) and final evaluation (B), N = 18.

Phosphate Binder Drug	Daily Dose (Number of Tablets per Day)	N (From a Total of 18 Individuals)	%
*Phosphate binder (after first and second evaluations)*			
OsvaRen	6	3	17%
Prodial	6	11	61%
Renagel	6	4	22%
Total		18	100%
*Phosphate binder (after the final evaluation)*			
OsvaRen	5	2	11%
OsvaRen	6	1	6%
Prodial	3	2	11%
Prodial	5	6	33%
Prodial	6	3	17%
Renagel	4	2	11%
Renagel	5	1	6%
Renagel	6	1	6%
Total		18	100%

## Data Availability

All data are available in the manuscript and Appendix A.

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
