# Peer review of "The Impact of Medical Nutrition Intervention on the Management of Hyperphosphatemia in Hemodialysis Patients with Stage 5 Chronic Kidney Disease: A Case Series"

_ijerph, 2023, doi:10.3390/ijerph20065049_

Round 1
Reviewer 1 Report
The study investigated the Impact of Medical Nutrition Intervention on the Management of Hyperphosphatemia in Hemodialysis Patients. Although the obtained results might have important clinical implication, some points need to be addressed, as follows:
1. In the method section, please explain the method of measuring biochemical variables such as phosphorus, calcium, PTH, etc.
2. In the results section, the findings of the principal component analysis are presented, while in the data analysis section, this analysis and its purpose are not mentioned. Please add some explanations about PCA in corresponding section.
3. It seems that some parts of table 2 are messed up. Please review and correct it.
4. In Figure 3, the results are shown in both table and graph form. It is suggested to present only one form (table or graph) to save page space and confusion for the readers.
5. In the present study, MNT was used together with phosphate binders to reduce phosphatemia, so it is suggested that in the conclusion section, lines 363-367, it should be mentioned that "following a low phosphorus diet reduces serum phosphorus and the received dose of phosphate binders”.
Author Response
Dear Reviewer 1,
We highly appreciate your efforts in reviewing our manuscript and the professional, accurate, and helpful comments to improve its quality; we responded to all your comments and made all suggested changes. Thank you for your excellent collaboration and support, and we wish you all the best.
You can find our detailed responses in the attachment, point by point.

Reviewer 2 Report
The main limitation of the study is the lack of a control group.
Other comments of this reviewer are:
- Provide the inclusion and exclusion criteria of the study.
- - Table 3 Phosphatemia is mentioned twice.
- Results and Discussion are in the same sector. You have to check if it is in accordance with journal’s policy.
- Please make more clear if the number/total dose of phosphate binders were reduced during the study. If this is the case then you should add it in the abstract as well.
- Maybe phosphate binder is a more appropriate term than phosphate chelator
Author Response
Dear Reviewer 2,
We highly appreciate your efforts in reviewing our manuscript and the professional, accurate, and helpful comments to improve its quality; we responded to all your comments and made all suggested changes. Thank you for your excellent collaboration and support, and we wish you all the best.
You can find all our detailed responses in the attached document.
